# Sparkling Cider Paired with Italian Cheese: Sensory Analysis and Consumer Assessment



**Giovanna Lomolino \***, **Matteo Marangon** [ID], **Simone Vincenzi** and **Alberto De Iseppi** [ID]

Department of Agronomy, Food, Natural Resources, Animal and Environment DAFNAE, University of Padua, Viale dell'Università 16, Legnaro, 35020 Padova, Italy
**\*** Correspondence: giovanna.lomolino@unipd.it; Tel.: +39-(0)-4982-72917; Fax: +39-(0)-4982-72919

**Abstract:** Cider is a beverage belonging to the tradition of many European rural areas. Pairing beverages and cheeses, even if it is part of an ancient tradition, is gaining more and more interest from the consumer. For this reason, in this research, we wanted to conduct a preliminary study on the combination of cider and cheese. In particular, six Italian sparkling ciders were selected, obtained through the Charmat and Champenoise method, and four types of Italian cheeses, from the Veneto region: *Casatella Trevigiana*, *Fienil*, *Morlacco* and *Ubriaco*, with very different sensory characteristics. The cider-cheese pairing test, conducted by a panel of experts, revealed how some cider parameters are reduced in intensity, such as astringency, while others are enhanced, such as fruitiness and persistence taste aroma. The hedonic test, conducted on the matching by 90 consumers, promoted some combinations while others were rejected. The sensory parameters associated with liking were fruity and taste aroma persistence, particularly expressed in some cider-cheese pairings.

**Keywords:** cider; cheese; food pairing; sensory analysis; consumer assessment

## 1. Introduction

Cider is a traditional beverage obtained from the alcoholic fermentation of apples, with alcohol content comprised between 2 and 8.5% and with a sour taste mostly due to the presence of malic acid [1]. Fermented products based on apples are very popular and appreciated in many European rural areas. In fact, cider is widespread in the United Kingdom, the largest consumer and producer in the world, in France (especially in Brittany and Normandy regions), Spain (particularly concentrated in the Asturias region), Germany, Ireland, the Netherlands, Finland, and Switzerland. In Italy, cider is less popular, and its consumption is generally mostly limited to Anglo-Saxon style pubs [2].

In the Italian market, the production and marketing of apple cider, or generally fermented fruit other than grapes, is not widespread. Some cider-growing realities are present on the Italian national territory but are essentially aimed at an artisanal production linked to ancient traditions.

As far as cider-making is concerned, the areas are concentrated in northern Italy, where cider has a tradition: Piedmont, Trentino Alto-Adige, Friuli Venezia Giulia and Val d'Aosta. The productions of these Italian regions, however, are more limited than those of French, especially in Normandy which holds the record, and of the Spanish or English ones.

To hinder its diffusion, there is above all the competition from wine, the most deeply rooted alcoholic beverage in Italy. Today, however, in Italy cider production companies are once again appearing on the market. Furthermore, this product could solve the problem of recovering certain mountain areas, enhancing the typical production which can constitute a good source of income in an agricultural economy. The search for a subtle balance between the typical organoleptic aspect, the production technologies and the high naturalness that distinguish this beverage could certainly favor the approach of potential consumers. [https://sidrodimele.it/ accessed on 27 November 2022].

The European Cider and Fruit Wine association (AICV) states that, in recent decades, there has been a growing interest in alcoholic beverages obtained from the fermentation of apples and other fruits, compared to other alcoholic beverages [1]. This growing popularity might be reflected on the number of scientific articles published in recent years on this topic. Generally, these studies mainly dealt with the characterization of the different aromatic compounds, the incidence of the apple variety and ripeness in conferring some distinctive characteristics to the finished product [1,3], and the different cider production technologies [3–7].

In this context, the recent increase in the demand for sparkling alcoholic beverages [8] has provided scope to develop new methods of cider production to meet the consumers' demands. Nowadays, besides the traditional production methods involving a bottle fermentations similar to the Champenoise method applied in the production of Champagne wine [9], the Charmat method is constantly becoming more popular in cider production [9]. An example of cider obtained with this traditional method is the "*Sidre de Asturias*", a product protected under the Spanish Designation of Origin rules (EEC Commission Regulation No 2154/2005) [10]. In this traditional production, the second alcoholic fermentation is conducted by adding yeast and sucrose to the base cider, so that fermentation and prolonged contact with lees take place in the same bottle that will be purchased by the consumer. During the lees ageing, the autolysis of the yeasts occurs, and this is responsible for the release of compounds that profoundly modify the organoleptic and foaming characteristics of the cider [9]. In particular, the main changes consist in the variation of the content in polysaccharides, in the nitrogen component and in the aromatic compounds [11,12].

Foam is one of the main characteristics of sparkling cider as it is the first attribute that the consumers perceive. The formation and stability of the foam represent the main characteristics that define the phenomenon. The presence of small bubbles that slowly rise through the liquid is highly appreciated, as is their persistence. Liger-Belair and colleagues [13] underline how gas bubbles are responsible for releasing and conveying aromatic components and imparting important sensory characteristics in the tasting of sparkling wines.

Cider can be consumed alone or as an alternative to wine and beer, in combination with various foods, considering different sensory characteristics and hedonic aspects. However, most of the studies on the hedonic response of consumers to food-beverage combinations concern research on food-wine combinations. However, in some social scenarios, the consumption of wine alone is considered inappropriate and therefore it is not often practiced, perhaps because this behavior is seen as socially unacceptable, but also because it is believed that consuming the wine together with food improves the social setting and the sensory experience in its entirety [14–16].

Much of the information on the theory of food-beverage combinations comes from wine and food writers, sommeliers and beverages and culinary experts, and is disclosed in popular press, on websites and culinary magazines. Unfortunately, scientific studies on foods-beverages pairing, in which the interactions between the chemical and structural components of drink and food are studied and controlled, are scarce [17]. This could be due to the difficulty in identifying the interactions and the food-beverage compounds responsible for the final sensory and hedonic result. For this reason, the study on combinations still represents a scientific challenge. On the other hand, as reported by Donadini and colleagues [18], (beer-cheese combination), pairing in general has a long history; already in the Middle Ages, beer was combined with various foods not only because they provided essential nutritional principles, but also because they were part of traditional recipes [19]. In fact, matching food and beverages is rooted in cultural practices. However, regardless of the traditional combination, a perfect match, with measurable characteristics, between beverages and cheeses requires a precise identification and selection of each element of the pair for the appreciation of the combination [18].

Many experts in the beverage (e.g., wine and beer) and dairy products industries and consumer experts have raised some interest in the theory of food-beverage pairing to

better understand which beverages are best paired with certain types of cheeses [20,21]. Italy offers a broad scenario for exploring beverages and cheeses as well as their sensorial relationships. In fact, as one of the biggest cheese producers in the world, it offers numerous types of products which are distinguished by texture, flavor, milk origin and fat content [22].

In Italy, undoubtedly, wine is the most popular alcoholic beverage. However, the consumption of other fermented drinks have always been present in the culture of the country. Among them, cider is experiencing growing interest from consumers, which are currently attracted to beverages with a moderate alcohol content.

Given the scarce scientific documentation on the cider-cheese combination, this research aims to study the sensory profile of six Italian sparkling ciders (obtained through both the Charmat and Champenoise method) and four types of Italian cheeses (produced from cow's milk) in order to explore which types of cheeses are best paired with sparkling ciders. These six Italian sparkling ciders were chosen because they are known in the production areas and marketed in Italy; moreover, although they belong to the sparkling category, they have different sensory characteristics, linked to the tradition (apple variety and cider production techniques) of the production area. Even if there are other areas and types of ciders in Italy, the study is preliminary and could represent the beginning for exploring the pairing of ciders with other typical cheese and/or foods. The study of the sensory profiles of cider alone and in combination with cheese will allow us to study the changes in perceptions induced on cider following the matching. Finally, 90 consumers will be asked to judge the combinations through a hedonic test.

## 2. Material and Methods

### 2.1. Cider

Six Italian sparkling ciders were selected for this study. The samples were the following:

- Maley Cidre Jorasses (M): Sidro Classico, classic method (Champenoise method), 7.5% alcohol.
- Maley Cidre du Mont Blanc Ancestral (MA): Sidro Ancestrale, (Champenoise method), 4.5% alcohol.
- Melchiori (Mel): Sidro di mela extra dry, (Charmat method), 8% alcohol.
- AsoloBio (AB): Sidro di mela, (Charmat method), 7.5% alcohol.
- Mosct secco (MS): (Champenoise method), 6.5% alcohol.
- Mosct dolce (MD): (Champenoise method), 4.5% alcohol.

### 2.2. Cider Sample Preparation for Sensory Analysis

The preparation of cider for sensory analysis was according to [9], with some modifications. The ciders were stored at 12 °C in the dark. During the sensory analysis, the cider samples were kept at 8 °C. The quantity of cider was 30 mL poured into a tulip shaped glass, covered with a small dish.

### 2.3. Cheese

Four cow milk cheeses were purchased from local farms and used for this tasting test. The choice was based on the seasoning period and production technique to maximize differences in flavors. The four selected cheeses represent typical Italian dairy products, belonging to the dairy tradition of the Veneto region.

*Casatella Trevigiana* PDO was chosen as fresh cheese, *Fienil* (3 months aging) and *Morlacco* (6 months aging) as medium-aged cheeses, and *Ubriaco* (12 months aging) as a mature cheese.

### 2.4. Cheese Samples Preparation for Sensory Analysis

The preparation of cheese for sensory analysis was conducted according to [23], with some modifications. Cheese samples were kept at room temperature (21 °C) for 30 min prior sensory analysis. Ten minutes prior to service, the rind of cheeses was removed (15 mm width), except for *Casatella Trevigiana*, that is fresh cheese without any rind. According

to the method of Donadini and colleagues [23], cheese samples were prepared by cutting them into 25 g strips (2.5 × 2.5 × 5.0 cm in size),labelled with a three-digit code in white paper dishes, and finally covered by a plastic film to preserve volatile compounds.

*2.5. QDA by Trained Panel*

Twelve trained panelists (seven females and five males) of 25–35 years old, were involved in sensory profiling of cider and cheese, by Quantitative Descriptive Analysis (QDA®). The test was carried out in a sensory analysis laboratory, at 21 °C and under artificial white light. The parameters considered along with the referenced used to train the panel are shown in Table 1 for cheese and in Table 2 for cider. Assessors rated each attribute using a discontinuous scale, from 0 to 10. The data were acquired by Fizz software (Biosystemes, Couternon, France). Tests were carried out on three different days, always in the morning, in different and distinct evaluation sessions both for cider and for cheese. The cheese samples were served in order of increasing taste-olfactory intensity. Therefore, the first was *Casatella Trevigiana*, followed by *Fienil*, *Morlacco*, and *Ubriaco*. On the other hand, ciders were randomly served without an intensity order of presentation as they all belonged to the same category (sparkling ciders).

**Table 1.** Parameters, their definitions used for descriptive analysis of cheese and standard material used during the panel training.

| CHEESE Sensory Descriptors | | |
|---|---|---|
| **Descriptors** | **Definition** | **Reference** |
| **Color** | The color of cheese, from white to yellow | Color scale from white to yellow, used for cheese |
| **Homogeneity** | The appearance of the cheese paste, whether it has irregularities, porosity or not. | Some examples of cheeses paste, from uniform to porous. |
| **Firmness** | The extent of resistance offered by cheese, assessed during the first five chews using the front teeth; ranging from soft to firm | Melted Emmental cheese, würstel, and carrot |
| **Vegetal** | The aromatic blend associated to grass, herbs, and vegetables | Grass, herbs, spinach |
| **Fruity** | The aromatic blend associated with different fresh fruits | Banana, apple, pear |
| **Fermented** | the pungent odor associated with fermented wort, or beer. | Weiss beer, half-fermented grape must |
| **Milk** | The aromatic smell associated to fresh milk | Cow milk |
| **Yogurt** | The aroma associated to plain yogurt | Fresh plain yogurt |
| **Almond** | The aroma associated to almond, and dry fruit | Almond, almond flour |
| **Salty** | The fundamental taste associated to NaCl | NaCl (0.5% $p/v$ in water) |
| **Bitter** | The fundamental taste associated to caffein | Caffein (0.1% in water) |
| **Sour** | The fundamental taste associated to acids | Citric acid (0.1% in water) |
| **Sweet** | The fundamental taste associated to sugar | Sucrose (5.0% in water) |
| **umami** | The fundamental taste associated to umami | Umami from Ajinomoto Company (0.5% in water) |
| **spicy** | The flavor associated to spices | Cloves, nutmeg, cinnamon |

The assessors had to wait 5 min between one sample to the other during which they were asked to eat unsalted crackers and rinse their mouth with water. Each sample (cheese or cider) was marked with a three-digit random number established before the test.

**Table 2.** Parameters, their definitions used for descriptive analysis of cider and standard material used during the panel training.

| CIDER Sensory Descriptors | | |
|---|---|---|
| **Descriptors** | **Definition** | **Reference** |
| **Foam persistence** | The time, in seconds, in which foam is present, until the total collapse | Is the foam persistence expressed in seconds, until all the bubbles collapse, when $CO_2$ injection is interrupted, in a model cider |
| **Color** | The shades of white wine, from white to straw yellow | Color scale from white to yellow, used for sparkling white wines |
| **Perlage** | The little chain bubbles that rise from the bottom of the glass to the surface | A Prosecco wine |
| **Apple** | Flavor of apples | Fresh apples (Granny Smith, Goldel, Stark) |
| **Fruity** | Flavor of banana, pear, pineapple | Fresh banana, pear, pineapple |
| **Honey** | Flavor reminiscent honey | Acacia honey |
| **Floreal** | Flavor reminiscent white flowers | Acacia, jasmine |
| **Spiciness** | Aroma reminiscent of spices | Benzyl-alcohol (10 mg/L in 5% water solution) |
| **Sulfurous** | The flavor of sulfurous (the typical smell of the match) | 1: 20 drops $SO_2$, 15 mL wine (Hood White 2015) |
| **Yeast** | The flavor of yeast | 0.25 g baker's yeast, 15 mL water. |
| **Smell intensity** | Defined as the total intensity of the perceived aroma | |
| **Bitter** | The fundamental taste perceived in the presence of caffeine | Caffeine (0.1% $p/v$ in 5% ethanol water solution) Donadini 2013 |
| **Sour** | The fundamental taste perceived in the presence of acids | Citric acid (0.1% in 5% ethanol water solution) |
| **Salty** | The fundamental taste perceived in the presence of salt | NaCl (0.5% $p/v$ in water) |
| **Sweet** | The fundamental taste perceived in the presence of sugar | Sucrose (5.0% $p/v$ in 5% ethanol water solution) |
| **Astringent** | Tactile perception that causes roughness in the oral cavity | Tannic acid (0.5% $p/v$ in 5% ethanol water solution) |
| **Umami** | The fundamental taste perceived in the presence of umami | monosodium glutamate |
| **Metallic** | Smell similar to rusty iron | Ferrous sulfate (3 mg/L in 5% ethanol water solution) |
| **Sparkling in the mouth** | Tactile perception, at the level of the oral cavity, generated by the carbon dioxide of the cider | Sparkling water |
| **Persistence taste aroma** | The time of persistence of taste and aroma in the mouth | |

*2.6. Sensory Cider Profile with and without Cheese Consumption*

For the cider-cheeses pairing test, a previously suggested method [23] was followed, with modifications related to this case study. The same trained panel, that assessed ciders and cheeses, was involved in tasting the cider-cheese combination. Six ciders and four cheeses gave rise to twenty-four combinations which were tested over six panel sessions. The analyzes were repeated in duplicate (one in the morning and one in the afternoon). In a first preliminary session, the descriptors that best defined the characteristics of the cider-cheese combination were identified. The descriptors chosen for this sensory test were effervescence perceived in the oral cavity, acid, bitter, salty, sweet, fruity, astringent,

odor intensity, persistence of taste and aroma. In a second session, the intensities of the descriptors were established. A scale from 0 to 10 was considered.

The procedure took place with the intake of the cider alone, after 5 min, they tasted the cheese as follows: (1) a bite of cheese, (2) chewing for 5 s, (3) drinking a sip of cider, (4) chew cheese and cider together for 5 s before ingesting. The judges noted the intensity of the descriptors after the combined tasting; 15 min elapsed between one taste and the next [23] and mineral water was drunk to rinse the mouth. The data were collected using Fizz Network sensory software (Biosystemes, Couternon, France) through a computer network, in computerized individual booths located in a sensory laboratory under controlled conditions as required in ISO 8589:2010.

### 2.7. Consumer Test

Ninety-six consumers were recruited from students and employees of the Agripolis Campus (University of Padua). The tasting took place inside the university premises. Consumers were aged between 19 and 60 (68% were between the ages of 19 and 27) and were made up of 59% males and 41% females. Before tasting, consumers were profiled by filling out a questionnaire on demographic data and on the frequency of consumption of cider, cheese and sparkling wines. The results of the questionnaire revealed that only 9% of consumers consumed cider; for the remaining consumers the tasting was not regular, while the consumption of sparkling wines reached the percentage of 88%. As for cheeses, consumers regularly ate cheese during the week (91%).

In a preliminary phase, consumers were instructed on the methods of separate tasting (cider and cheese) and combined (cider-cheese, together). The consumer was asked to (1) taste a bite of cheese, (2) chew it for 5 s, (3) drink a sip of cider and (4) chew cheese and cider together for 5 s. Between one taste and the next, people were asked to wait 10 min during which they were asked to rinse their mouth with mineral water. The 24 cheese/cider combinations were served randomly and numbered with three-digit numbers. The preference was evaluated on a hedonic scale from 0 (extremely dislike) to 10 (extremely like).

The consumer test was organized over 8 days (four cider/cheese combinations each), in order not to tire the consumers, and each session lasted about 60 min.

### 2.8. Statistical Analysis

Data were statistically processed by Excel (Microsoft Corporation, Redmond, WA, USA), Statgraphics Centurion XVI (StatPoint Technologies Inc., Warrenton, VA, USA) and OriginPro (OriginLab Corporation, Northampton, MA, USA). A descriptive statistical study, spider plots (sensory parameters), was conducted. Analysis of variance (ANOVA) followed by Tukey's test and HSD were applied for inferential study ($p < 0.05$). Principal component analysis (PCA) was applied to identify sensory parameters, ciders and cider-cheese paired.

## 3. Result and Discussion

### 3.1. Sensory Analysis of Cheese by Trained Panel

Table 3 shows the sensory profile of the four cheeses used in the pairing tests with cider.

The types of cheese used are very different from each other; in fact, the aim is to explore how different characteristics of cheese could be paired with those of ciders.

As shown in Table 3, the four samples show very different characteristics. In fact, *Casatella,* a fresh cheese, reported a milky white color with a very homogeneous paste and a soft consistency. The prevalent aroma is milk, but fermented and yogurt smell are also present, while the aromas related to seasoning (almond) are almost absent. The taste is rather sweet, while the other flavors are of very low intensity (see mean value of Table 3 for *Casatella* cheese) and this highlights the delicate character of this cheese, which presents, as a prevailing note, the milky characteristic. The sensory profile found in the *Casatella* is typical of fresh cheese with a high-water content. In these products, the low proteolysis and lipolysis does not allow for the release of odors typical of aged cheeses [24].

**Table 3.** Quantitative Descriptive Analysis (QDA) of cheeses. Values are the mean of 12 trained panelists; tests were carried out in three different days (three replicates). Different letters in the rows are significantly different at *p* < 0.05 according to Tuckey test.

| | *Casatella Trevigiana* | *Fienil* | *Morlacco* | *Ubriaco* |
|---|---|---|---|---|
| **Color** | 1.7 ± 0.8 [c] | 7.3 ± 0.8 [a] | 5.2 ± 0.8 [b] | 7.8 ± 0.9 [a] |
| **Homogeneity** | 8.3 ± 0.7 [a] | 7.3 ± 0.6 [ab] | 6.8 ± 0.5 [b] | 9.5 ± 0.8 [a] |
| **Firmness** | 2.7 ± 0.7 [c] | 6.9 ± 0.8 [b] | 5.7 ± 0.8 [b] | 8.6 ± 0.9 [a] |
| **Vegetal** | 4.2 ± 0.6 [b] | 6.4 ± 0.5 [a] | 7.0 ± 0.7 [a] | 4.2 ± 0.6 [b] |
| **Fruity** | 3.4 ± 0.6 [b] | 3.1 ± 0.9 [b] | 2.7 ± 0.8 [b] | 7.8 ± 1.1 [a] |
| **Fermented** | 7.7 ± 0.8 [a] | 3.8 ± 0.7 [c] | 5.8 ± 0.6 [b] | 6.6 ± 0.7 [b] |
| **Milk** | 8.8 ± 1.1 [a] | 4.8 ± 0.9 [c] | 6.9 ± 0.7 [b] | 6.8 ± 1.0 [b] |
| **Yogurt** | 5.4 ± 0.9 [b] | 2.7 ± 0.7 [c] | 7.2 ± 0.6 [a] | 2.6 ± 0.8 [c] |
| **Almond** | 1.7 ± 0.8 [b] | 5.6 ± 0.9 [a] | 5.5 ± 1.0 [a] | 5.2 ± 1.0 [a] |
| **Salty** | 2.9 ± 0.5 [c] | 4.5 ± 0.7 [b] | 7.2 ± 0.8 [a] | 7.7 ± 1.1 [a] |
| **Bitter** | 1.3 ± 0.7 [d] | 4.1 ± 0.4 [c] | 6.4 ± 0.6 [a] | 5.1 ± 0.5 [b] |
| **Sour** | 2.1 ± 0.9 [c] | 1.9 ± 1 [c] | 5.8 ± 0.7 [a] | 4.0 ± 0.6 [b] |
| **Sweet** | 3.8 ± 0.5 [a] | 1.9 ± 0.7 [ab] | 2.7 ± 0.9 [a] | 3.5 ± 0.9 [a] |
| **Umami** | 2.7 ± 0.8 [b] | 5.7 ± 0.7 [ab] | 6.6 ± 0.7 [a] | 7.6 ± 1 [a] |
| **Almond** | 1.7 ± 0.7 [c] | 2.1 ± 1.0 [c] | 6.7 ± 0.6 [a] | 4.8 ± 0.6 [b] |
| **Fermented** | 2.5 ± 1.0 [b] | 1.9 ± 0.8 [b] | 7.3 ± 0.6 [a] | 2.4 ± 0.7 [b] |
| **Spicy** | 1.0 ± 0.0 [c] | 5.5 ± 0.3 [b] | 6.9 ± 0.4 [a] | 7.5 ± 0.7 [a] |
| **Milky** | 7.6 ± 0.9 [a] | 5.8 ± 0.5 [ab] | 6.8 ± 1.0 [a] | 5.6 ± 0.8 [ab] |

*Fienil* (see Table 3) has characteristics typical of an intermediate seasoning cheese; the aroma is reminiscent of vegetables, due to the presence of straw used during aging, in addition to the note of almond. On the palate, *Fienil* is slightly salty, has a note of umami, and a milky and spicy flavor. In summary, *Fienil* reported sensory characteristics that are placed between fresh cheese and medium-aged cheese. *Morlacco* is a whitish cheese (Table 3), the paste is less homogeneous than the others and rather firm. It has vegetal, milky, fermented yoghurt, and almond smells. The taste is sapid, salty, slightly bitter, acid, and umami. The flavor is also pronounced, with very pronounced notes of almond, fermented, milk and spicy. Finally, the *Ubriaco* (Table 3) has a yellowish color, the paste is homogeneous and hard, typical of aged cheese. The predominant fruity aromatic note could be attributed to the presence of grape pomace, with which the cheese is covered during aging. It also has an almond smell. *Ubriaco* is a savory cheese, with a salty and umami taste, slightly bitter. The flavor is characterized by spicy and milky notes (see Table 3, mean values of sensory parameters).

Apart from the aging time, the production technology (the use of straw and grape pomace for *Fienil* and *Ubriaco*, respectively), gave the cheeses unique and distinctive characteristics. The choice of such different samples is useful in identifying those characteristics of the cheese that prevail or not in the subsequent pairing phases [18].

### 3.2. Sensory Analysis of Cider by Trained Panel

Compared to wine making, the complexities of cider making have been long underestimated. With the recent introduction of techniques derived from the oenological industry in the production of ciders, we can witness a diversification of products characterized by

high quality standards in terms of flavor and, in the case of sparkling ciders, *perlage* and effervescence [1].

　　Foam is one of the most important characteristics of sparkling ciders, because bubbles are the first parameter that the consumer perceives. In fact, the first characteristics that are appreciated in a sparkling cider are the initial foam, the persistence, and the size of the bubbles [9].

　　The ciders analyzed (Table 4), although they all belong to the "sparkling" category, each have sensory characteristics that distinguish them. The ciders (see also materials and methods), purchased from the local farms, were obtained through the Champenoise (M, MA, MS and MD) and Charmat methods (Mel, AB). Maley Classico (M) (Table 4), which at first sight present intermediate characteristics of foam persistence, color and *perlage*.

**Table 4.** Quantitative Descriptive Analysis (QDA) of ciders. Values are the mean of 12 trained panelists; tests were carried out in three different days (three replicates). Different letters in the rows are significantly different at $p < 0.05$ according to Tuckey test.

| | Maley Classico | Maley Ancestrale | Melchiori | Asolo Bio | Mosct Secco | Mosct Dolce |
|---|---|---|---|---|---|---|
| **Foam persistence** | 3.9 ± 0.9 [b] | 7.6 ± 0.9 [a] | 4.4 ± 0.8 [b] | 6.7 ± 0.8 [a] | 5.5 ± 0.7 [ab] | 2.7 ± 0.3 [c] |
| **Color** | 4.7 ± 0.4 [b] | 3.7 ± 0.5 [b] | 4.4 ± 0.3 [b] | 2.1 ± 0.2 [c] | 2.2 ± 0.1 [c] | 7.0 ± 0.2 [a] |
| **Perlage** | 4.8 ± 0.8 [b] | 7.4 ± 0.9 [a] | 5.1 ± 0.7 [b] | 6.1 ± 0.5 [b] | 5.6 ± 0.5 [b] | 3.0 ± 0.5 [c] |
| **Apple** | 7.9 ± 1.4 [a] | 2.7 ± 0.7 [c] | 5.0 ± 0.9 [b] | 5.9 ± 0.7 [b] | 3.5 ± 0.8 [c] | 8.3 ± 0.8 [a] |
| **Fruity** | 7.6 ± 1.3 [a] | 3.9 ± 0.7 [b] | 3.9 ± 0.7 [b] | 6.6 ± 0.6 [a] | 3.7 ± 0.6 [b] | 6.4 ± 0.8 [a] |
| **Honey** | 4.6 ± 0.7 [a] | 2.6 ± 0.6 [b] | 4.8 ± 0.9 [a] | 2.6 ± 0.9 [b] | 1.6 ± 0.6 [b] | 5.2 ± 0.9 [a] |
| **Floral** | 4.7 ± 0.8 [a] | 3.7 ± 0.8 [a] | 2.7 ± 0.7 [ab] | 3.6 ± 0.8 [a] | 2.8 ± 0.8 [ab] | 4.5 ± 0.6 [a] |
| **Spicy** | 4.9 ± 0.9 [b] | 3.1 ± 0.8 [c] | 1.6 ± 0.8 [c] | 2.5 ± 0.7 [c] | 7.1 ± 0.6 [a] | 4.5 ± 0.5 [b] |
| **Sulfurous** | 1.1 ± 0.3 [c] | 2.8 ± 0.6 [b] | 5.4 ± 0.6 [a] | 1.3 ± 0.6 [c] | 1.4 ± 0.5 [c] | 1.1 ± 0.2 [c] |
| **Yeast** | 1.5 ± 0.6 [bc] | 2.5 ± 0.8 [b] | 5.6 ± 0.6 [a] | 1.8 ± 0.7 [bc] | 3.1 ± 0.6 [b] | 1.9 ± 0.6 [bc] |
| **Smell intensity** | 6.9 ± 1.2 [a] | 6.7 ± 0.5 [a] | 5.6 ± 0.7 [ab] | 5.8 ± 0.4 [ab] | 7.6 ± 0.7 [a] | 5.8 ± 0.9 [ab] |
| **Bitter** | 1.7 ± 0.7 [c] | 4.8 ± 0.7 [b] | 6.2 ± 0.5 [a] | 3.7 ± 0.8 [b] | 4.4 ± 0.6 [b] | 1.3 ± 0.7 [c] |
| **Sour** | 4.3 ± 0.8 [a] | 3.4 ± 0.8 [ab] | 4.9 ± 0.8 [a] | 5.7 ± 0.8 [a] | 4.1 ± 0.8 [a] | 1.7 ± 0.5 [b] |
| **Salty** | 2.1 ± 0.8 [a] | 1.6 ± 0.6 [ab] | 2.9 ± 0.7 [a] | 3.3 ± 0.7 [a] | 1.2 ± 0.5 [ab] | 2.2 ± 0.7 [a] |
| **Sweet** | 5.0 ± 0.8 [b] | 2.6 ± 0.7 [c] | 2.2 ± 0.8 [c] | 3.1 ± 0.6 [c] | 1.3 ± 0.4 [d] | 7.6 ± 1.0 [a] |
| **Astringent** | 4.4 ± 0.9 [ab] | 3.5 ± 0.8 [b] | 3.8 ± 0.4 [b] | 5.4 ± 0.7 [a] | 1.3 ± 0.5 [c] | 6.2 ± 0.9 [a] |
| **Umami** | 1.7 ± 0.6 [ab] | 2.6 ± 0.6 [a] | 2.1 ± 0.7 [a] | 3.5 ± 0.9 [a] | 1.3 ± 0.3 [b] | 3.1 ± 1.0 [a] |
| **Apple** | 8.0 ± 0.9 [a] | 3.2 ± 0.6 [c] | 4.8 ± 0.8 [b] | 5.6 ± 0.6 [b] | 2.9 ± 1.0 [c] | 8.1 ± 0.9 [a] |
| **Fruity** | 6.6 ± 0.9 [a] | 2.7 ± 0.7 [b] | 3.2 ± 0.7 [b] | 6.1 ± 0.6 [a] | 4.1 ± 0.6 [b] | 5.9 ± 0.8 [a] |
| **Metallic** | 1.4 ± 0.9 [b] | 4.1 ± 0.8 [a] | 2.0 ± 0.4 [b] | 3.9 ± 0.7 [a] | 1.3 ± 0.5 [b] | 1.2 ± 0.9 [b] |
| **Sparkling** | 6.1 ± 0.8 [a] | 6.7 ± 0.8 [a] | 6.3 ± 0.7 [a] | 7.2 ± 0.5 [a] | 4.3 ± 0.6 [b] | 3.6 ± 0.8 [b] |
| **Persistence taste aroma** | 6.4 ± 1.0 [a] | 7.1 ± 0.8 [a] | 7.2 ± 0.8 [a] | 6.3 ± 0.7 [a] | 5.8 ± 0.9 [a] | 5.3 ± 0.5 [ab] |

　　What characterizes this cider is the apple aroma, the fruitiness, the hint of honey, the floral and, in general, the olfactory intensity. To the taste, it has a certain acidity and astringency, while in the flavor the scent of apples and fruit reappears; on the palate, the tactile perception of the effervescence is quite strong compared to other ciders.

　　Maley Ancestrale (MA) cider, obtained with the Champenoise method, (Table 4), presents, to the sight, persistent foam and a lasting *perlage*; at the olfactory it does not have an accentuated apple smell but on the other hand the floral and slightly fruity note gives the

cider a certain olfactory intensity. On the palate, it is a little sour, salty and umami, perhaps because of the long contact the cider had with the yeast lees, typical of the Champenoise method [25]. The MA finally presents the perceptions of astringency and metallic. Similarly to all other ciders, it does not stand out for its persistence taste and aroma which is high in all cases.

Melchiori cider (Mel), produced with the Charmat method, (Table 4), at sight presents intermediate characteristics relating to the characters "foam persistence", "color" and "*perlage*"; however, what distinguishes it is the note of honey, sulfurous and yeast responsible for the aromatic intensity. On the palate it is bitter, acidic and slightly salty as well as having an umami taste note; moreover, the strong effervescence in the mouth is also a characteristic of this cider and typical of wine produced with Charmat method in which, comparted to the Champenoise one, a higher pressure inside the bottle is generally reached [26].

Asolo Bio (AB) cider (Table 4), produced with the Charmat method, stands out for its persistence foam and an intermediate *perlage*. The aroma is fruity and could be attributable to apple, it also has notes of honey, but also sulfur and yeast perceived with a certain intensity. On the palate, it is sapid thanks to the characteristics of bitterness, acidity, salty and umami taste as well as having a perceptible sparkling in the mouth. Mosct Secco (MS) cider (Table 4), obtained with the Champenoise method, has, in general, intermediate characteristics compared to other ciders. In particular, the foam is quite persistent, but what distinguishes it is the spicy aroma which also gives it smell intensity and an acidic taste. The other sensory characteristics, even if present, are less pronounced than the other ciders.

Mosct Dolce cider, obtained with the Champenoise method (Table 4), stands out for being sweet and with a low alcohol content (4.5%). The color tends towards yellow and is characterized by aromatic notes of apple, fruity, honey and floral, characteristics that give the product its aromatic intensity. On the palate it is sweet, astringent and umami. The flavor of apple and fruit in general reappears in the mouth. The effervescence, even if present, is lower than the other ciders.

The sensory differences found among the six ciders can be mainly attributed to the alcoholic fermentations and the following contact with yeasts. In fact, the yeast cells are known to release different compounds such as lipids, amino acids, mannoproteins and enzymes are involved in the perception of flavors and in the stabilization of the foam [27]. Therefore, the choice of the yeast strain, the fermentation conditions and the contact time [28] are considered the main factors that influenced the sensorial differences which emerged among the tested ciders.

*3.3. Sensory Profiles of Ciders before and after Tasting the Cheeses*

The ciders were evaluated in combination with the cheeses. For this purpose, the nine descriptors which best identified ciders (Figure 1) were assessed before and after the combination with the various cheeses. As shown in Figure 1, the ciders, tasted alone (black profiles) and in combination, have different and characteristic profiles depending on the case study.

For example, some descriptors, such as "persistence taste aroma, smell intensity and salty" increase in intensity after tasting the cheeses, others remain unchanged or attenuated, depending on the case. In particular, (Figure 1) the Maley Classico (M) cider presents in the mouth, after tasting the cheese, the most attenuated sparkling character, as well as astringent, while the fruitiness increases in the presence of only the *Ubriaco* cheese.

When Maley Ancestral (MA) cider is considered, the effervescence is attenuated, the sweetness is enhanced, the bitterness decreases, and the fruitiness increases in intensity only when it is paired with *Ubriaco* and *Fienil* cheeses.

After tasting the cheeses, Melchiori (Mel) cider, shows modified profiles similar to the MA sample despite that the "fruity" descriptor is significantly higher in all cases. On the contrary, Asolo Bio (AB) after tasting the cheeses was perceived as less sparkling, sour, bitter, and astringent. The sweet descriptor is an exception, which is exalted in the presence of *Casatella*.

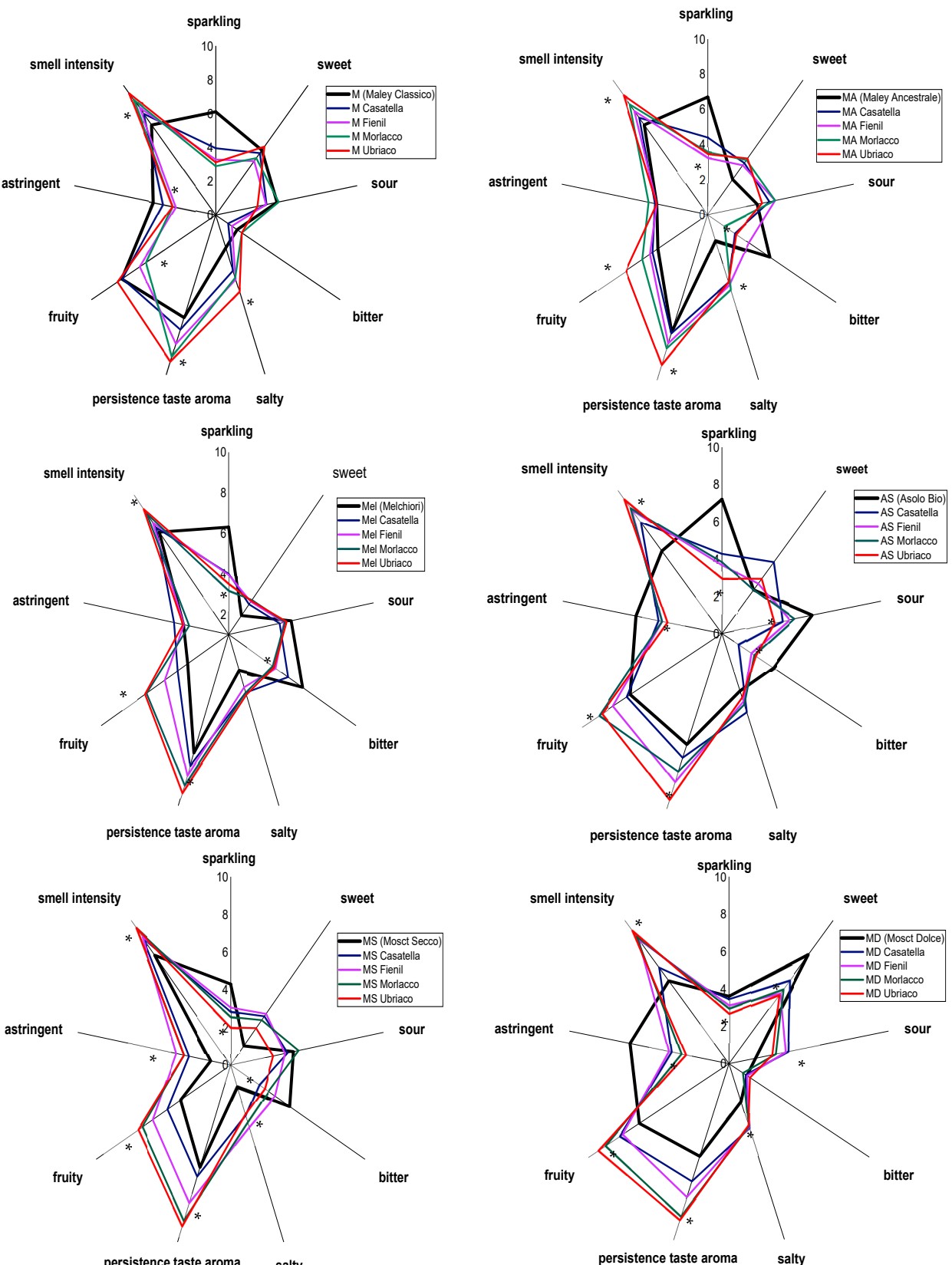

**Figure 1.** Quantitative Descriptive analysis (QDA), Spider plot of the ciders before and after tasting the cheeses, as reported in the legends of each graph. Values are the mean of 12 trained panelists; tests were carried out in three replicates. The asterisks (*) indicate significantly different means at *p* < 0.05 according to Tuckey test.

If the profiles of Mosct Secco (MS) are analyzed, sparkling and bitterness parameters were attenuated by the combination with cheeses. Conversely, astringency, sweet and fruity perception are enhanced.

Mosct Dolce deserves a separate consideration. In this case, due to the prevalence of sweet taste and the less perception of the bubbles (which did not vary after the cheeses tasting), the sweet perception decreases as does astringency, while the perceptions of acidity and fruitiness increase.

These results are in agreement with previous studies, carried out on wine-cheese [14] and beer-cheese pairing [23], where wine or beer tasting after cheese modify many of the descriptors of the beverage. According to Donadini et al. (2013) [23], who conducted beer-cheese pairing studies, the presence of cheese in the sensory evaluation of cider could bring important changes. The decrease/increase in some cider descriptors may be due to the masking or strengthening effect of some components of the cheese on those of the cider [23]. As previously reported [23], cheese could increase salty and umami perception, responsible for reducing bitter perception, as observed in several cider-cheese combinations (Figure 1). The reduction in astringency is observable in those cases where the starting cider has this highly expressed parameter. In these cases (Figure 1), the proteins of the cheese could compete with salivary ones in the binding of the cider's polyphenols thus determining a significant reduction in the perceived astringency [23]. This is in agreement with what observed in the case of M, AB, and MD (Figure 1). Furthermore, the fats present in the cheese attenuate the effervescence in all cases, as also observed by Donadini and colleagues [23]. Finally, contrary to what was observed by Bastian et al. [14], the persistence taste aroma and the smell intensity were increased in almost all cases; the question remains whether this effect is due to an enhancement of perception or is due to the strong odor component of the cheeses which in many cases dominates that of the ciders.

### 3.4. Assessment of Ciders and Pairing with Cheeses by Consumer

The ciders and the combinations with cheeses were subjected to evaluation by consumers, which expressed their overall liking on a scale from 0 to 10.

Before cheese tasting, the cider that received the lowest score was MS, while the most appreciated were MD, AB, and MA. Intermediate results were reported for Mel and M samples (Figure 2). After tasting with cheeses, the perception of ciders was different depending on the combined cheese. In particular, the overall liking of cider M increased in combination with *Casatella* and *Ubriaco* while it remained stable when the cider was paired with *Morlacco* and *Fienil*. A very different situation is shown for MA and Mel ciders. In fact, when they are paired with *Casatella* and *Fienil*, the consumers rejected the association. Conversely, when tested in combination with more aged cheeses (*Morlacco* and *Ubriaco*) the scores increased compared to ciders alone. Moreover, the score of MS decreases when it is combined with fresher cheeses (in this case including *Morlacco*), while it is enhanced by the pairing with the more aged *Ubriaco*.

An opposite trend was shown by the AB cider which score decreased with the increase in the cheese's aging period. Finally, in the case of MD cider, the high score reported when consumed alone is reproduced only by its combination with *Ubriaco* while the score decreased in the other cases, particularly in the case of the pairing with *Morlacco*.

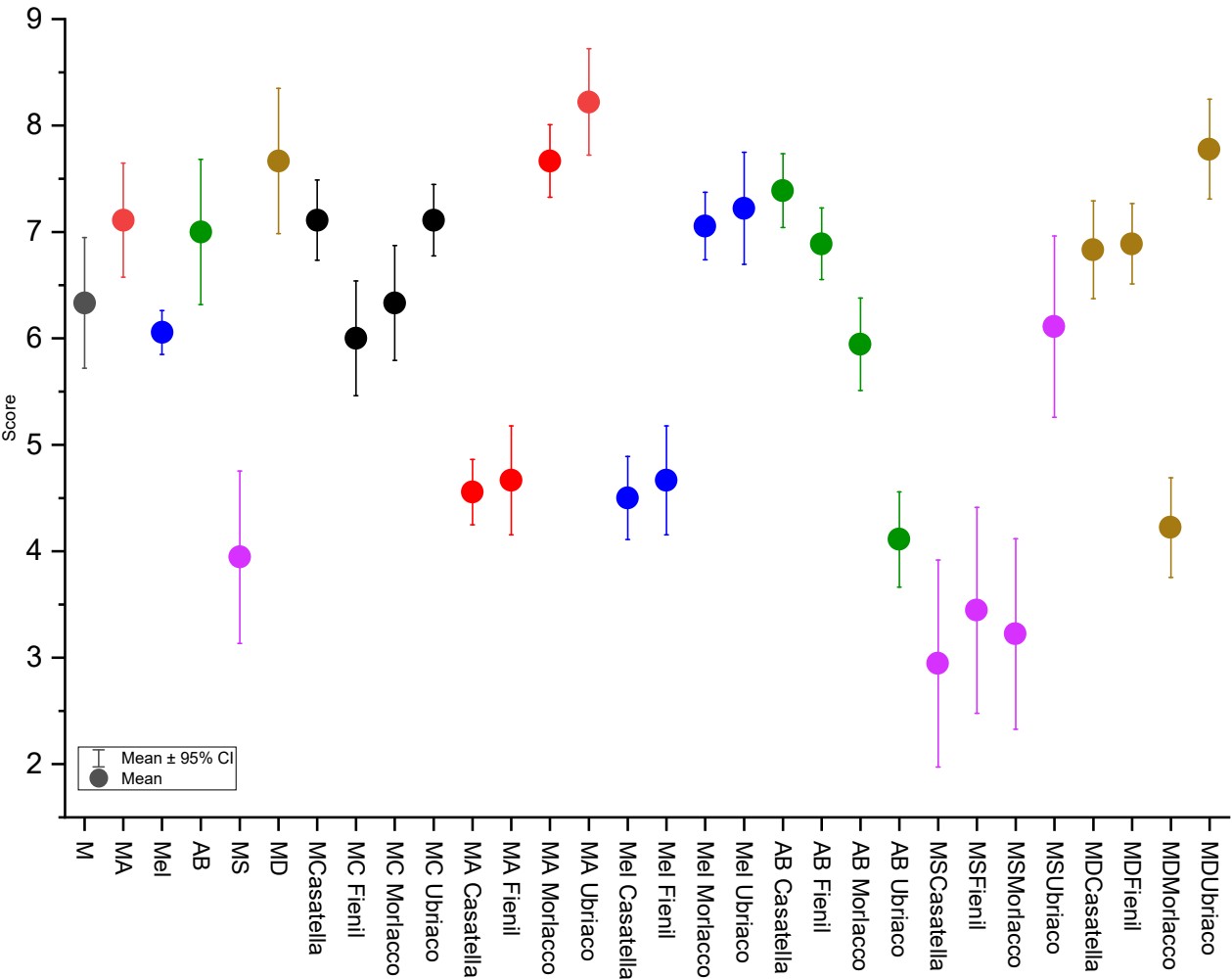

**Figure 2.** Consumer preference test. The preferences, with a score from 1 to 10, of the ciders, assessed alone, and after tasting the cheeses, were compared. Means and 95% confidence intervals were calculated.

### 3.5. PCA Ciders Alone and Paired with Cheeses

In order to verify the similarities regarding sensory characteristics, the PCAs of the ciders alone and paired with the different cheeses are reported in the four panels of Figure 3. When the ciders were paired with *Casatella* (Figure 3A), the two dimensions explain 82.06% of the total variance. The PC1 axis distinguishes, in the positive part, all the ciders alone and in combination with the *Casatella*. The PC2 axis divides the Mel, MS, MA ciders alone and in combination; moreover, it also includes AB. The negative part of PC2, on the other hand, includes the AB paired with *Casatella*, along with MD, M ciders alone and in combination with *Casatella*. The positive part of PC2 shows the ciders and combinations characterized by smell intensity, persistence taste aroma and sparkling, while in the negative part of PC2 the ciders and the combination with *Casatella* are characterized by a fruity perception and a better judgment of acceptability.

Figure 3B shows how the two dimensions account for 81.92% of the total variance. In the positive semi-quadrant of PC1, there are all the single ciders and in combination with *Fienil*. On the other hand, PC2, positive dial, presents the MD, M ciders alone and in combination with *Fienil*, in addition to the combinations MS and AB with cheese, all characterized by the fruity characteristic. On the contrary, the negative quadrant of includes Mel, MS, MA and AB alone and the combinations Mel and MA with *Fienil*, characterized by the persistence of taste aroma and smell intensity. The assessment does not reward combinations with *Fienil* (Figure 2; Figure 3B).

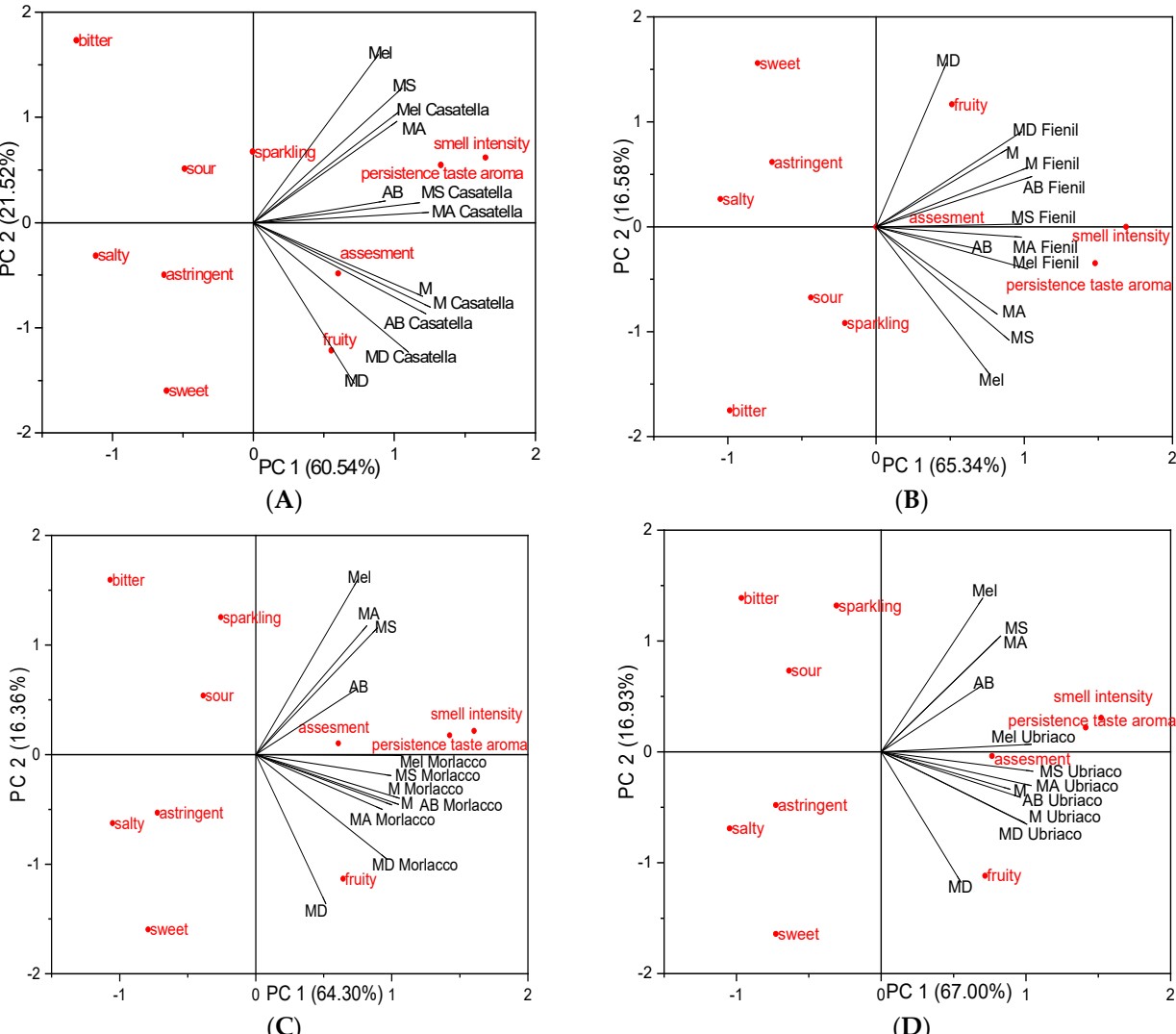

**Figure 3.** Principal Component analysis (biplot, PC1 vs. PC2) of the average descriptive data and consumer assessment of ciders before and after cheese tasting. (**A**): ciders before and after *Casatella* cheese; (**B**): ciders before and after *Fienil* cheese; (**C**): ciders before and after *Morlacco* cheese; (**D**): ciders before and after *Morlacco* cheese.

Figure 3C shows PCA of the individual ciders and in combination with *Morlacco* and indicates how the two dimensions account for 80.66% of the variance. Furthermore, in this case, the positive semi-quadrant PC1 presents all the ciders and their combinations with *Morlacco*; while for PC2, the positive semi-quadrant, shows the ciders alone Mel, MA, MS and AB associated with assessment, smell intensity and persistence taste aroma. The perception of the bubbles and sourness could also be associated with the positive field. In the negative field, there are the MD and M ciders and all the associations. The characteristic that distinguishes these combinations and the two ciders is the fruity perception. Figure 3D shows the PCA of the cider-*Ubriaco* pairings; the two dimensions explain 83.39% of the variance. The positive part of PC1 shows all the ciders and all the combinations with the *Ubriaco* cheese, all characterized by smell intensity and persistence taste aroma. All combinations are rewarded by positive judgment (assessment) and fruity scent, along with unmatched MD. PC2 shows, in the positive part, the four single ciders Mel, MS, MA and AB, characterized by the perception of sparkling, while Mel with *Ubriaco*, for the characteristics smell intensity and persistence smell aroma. All other combinations are rewarded by the assessment and the fruity note.

Data from Figure 3 generally highlights that the modification of the sensory profile can be mostly related to the degree of cheese aging. In particular, clear differences in cider perception are detectable between the datasets related to the fresher cheeses (*Casatella* and *Fienil*, Figure 3A,B) and the ones in which the more aged cheese (*Morlacco* and *Ubriaco*, Figure 3C,D) were paired with the various ciders. In the former case, each cider "alone" sample appear close to its analog "paired" with cheese, thus indicating a low impact of cheese in the modification of all the perceptions evaluated by the panels. This appears in accordance with Figure 2 in which, except for the Mel sample, the panel gave a similar score among "alone" and "paired" samples. Conversely, the pairing with *Morlacco* and *Ubriaco* seemed to have a higher impact on the sensory profile, improving the "fruity", "persistence", and "overall liking" (assessment) of the various ciders. This can be attributed to the higher sapidity, and the umami taste developed during aging that could have had a greater impact in modifying the sensory profile rather than the milky and sweet notes provided by the fresher *Casatella* and *Fienil* which are more in line with the sensory profile of the ciders tasted "alone". An exception to this explanation is represented by the MD cider which, being already in the IV quadrant (Figure 3C,D) when tasted alone, do not see its sensory profile much modified by the pairing with *Morlacco* and *Ubriaco*.

Since there are no previous studies on cider-cheese pairing, through the obtained results it can be stated that the consumption of ciders and combinations with cheeses has a great impact on the perceived smell intensity, persistence of taste and aroma, and fruitiness of the ciders. The bitter, sweet, and astringent parameters always appear opposite to the assessment. As reported by Bastian et al. [14], the modification of some wine attributes, caused by the previous tasting of cheddar cheese, can change the final assessment level of the pairing. This suggests that characteristics disliked in cider can be moderated by prior food consumption and result in an enhanced preference for the cider. The pairing assessment could be due to the exaltation of some characters and the suppression of less desirable ones, as in the case of the cider-cheese combination, shown here.

## 4. Conclusions

The results presented in this research on cider-cheese pairing are in all respects preliminary; for this reason, where possible, they have been compared with research conducted on combinations of other beverages and cheese. However, what emerged from this sensory experimentation highlights that the consumption of cheese before cider significantly changes the sensory profile of the cider itself. In some cases, the sensory parameters of the cheese reduce the perceptions of some attributes of the cider, as in the case of astringency; in others, they can increase them as in the case of the perception of fruitiness.

In particular, the reduction in the perception of astringency, after tasted the cheeses, is more evident in the ciders in which this parameter is more expressed; moreover, astringency and bitterness are always opposite to the assessment. This means that disliked characteristics in the cider can be modulated in the pairing, and result in an enhanced preference for the cider.

If the effervescence is considered, it is attenuated by the fat component of the cheeses, in all cases.

With regards to the olfactory intensity, it is not clear whether the parameter is due to an enhancement effect linked to the pairing or to the strong odor component of the cheeses, which in many cases dominates that of the ciders. In general, the sapidity and umami taste, developed during the aging of the cheeses, may have influenced the sensory profile of the pairings compared to the notes of milk and sweetness, typical of fresh cheeses, which are more similar to the sensory profile of cider, tested alone.

In any case, this study shows that cheese can be a complementary partner of cider and from the preliminary results it can be asserted that the parameters related to liking in the cider-cheese combination could be the fruity perception, smell intensity, taste and aroma persistance. The liking judgment of the pairing is often due to the type of cheese, in some cases the combination is appreciated, in others it is rejected, so it is necessary to select the

cider and the cheese in their pairing and understand the sensory parameters that promote the result.

**Author Contributions:** Conceptualization, G.L. and A.D.I.; methodology, S.V.; validation, M.M. and A.D.I.; formal analysis, G.L. and A.D.I.; investigation, M.M. and S.V.; resources, G.L.; data curation, G.L. and A.D.I.; writing—original draft preparation, G.L., A.D.I. and M.M.; writing—review and editing, G.L., A.D.I., M.M. and S.V.; supervision, S.V., M.M. and A.D.I.; funding acquisition, G.L. All authors have read and agreed to the published version of the manuscript.

**Funding:** This research was supported by University of Padova, Italy, with the funding DOR1847072 (2018) and BIRD165379 (2016).

**Data Availability Statement:** Data is contained within the article.

**Acknowledgments:** The authors thank the sensory panel for their work on this project.

**Conflicts of Interest:** The authors declare no conflict of interest.

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
