# Peer review of "Sparkling Cider Paired with Italian Cheese: Sensory Analysis and Consumer Assessment"

_beverages, doi:10.3390/beverages8040082_

Round 1
Reviewer 1 Report
All text must be technically edited according to the instructions. For example, table 1 (not bold) table 2 (bold). Font in the table 1 and 2? Table 3 - some unnecessary lines are present. Table 4 - the name is below the table?
Also, two samples have lower alcohol concentration and 4 samples have higher alcohol concentration. No analysis of the results was done with respect to this data. Can the results be further analyzed with respect to this parameter to perhaps reach additional conclusions?
Author Response
Dear Reviewer
We have added the corrections suggested, as follow.
Comments and Suggestions for Authors
All text must be technically edited according to the instructions. For example, table 1 (not bold) table 2 (bold). Font in the table 1 and 2? Table 3 - some unnecessary lines are present. Table 4 - the name is below the table?
As suggested, we have corrected in the text.
Also, two samples have lower alcohol concentration and 4 samples have higher alcohol concentration. No analysis of the results was done with respect to this data. Can the results be further analyzed with respect to this parameter to perhaps reach additional conclusions?
As suggested by the reviewer, the concentration of ethanol in general in alcoholic beverages could modify the sensory perception of the product. In this research, 2 of the 6 ciders considered have a low concentration of ethanol, equal to 4.5% and, for this reason, could have sensory characteristics affected by a lesser presence of alcohol. Indeed, as reported in the literature (King et al, 2013), the concentration of ethanol can affect the sensory perception of alcoholic beverages, as follows:
1- increases the perception of bitterness;
2- decreases the acid;
3- it could alter the perception of sweet;
4- could mask the perception of fruitiness;
5- reduces astringency.
The results obtained from the sensorial analysis of the ciders, tasted not paired, however, do not seem to highlight the listed effects, due to the alcohol concentration. Evidently, in this research the variables are multiple and can act on the perception of ciders in a way that is difficult to correlate, and/or, in some way, to justify. For this reason, the authors of the paper believe not to add discussion of the effect of alcohol on the sensory perception of ciders. Finally, the matching makes the role of alcohol on the modification of sensorial perception even more complex. We believe that, in order to study the effect of alcohol, another experimental design should be introduced with ciders and cheeses. However, if you think, we could add to this paper as follows:
“The ethanol content could affect the sensory parameters of the cider, as it happens in wine (King et al 2013). In this research, two ciders have a lower ethanol content than the others (4.5%) and it is predictable that a different concentration may affect the parameters observed when the beverage is tasted alone or in combination. However, the results obtained, while showing evident differences between the theses, do not justify the effect of ethanol on the other parameters, perhaps due to the numerous variables present in this research."
This is the possible explanation, however, we did not add the sentence into the text.
Reference: Ellena S. King, Randall L. Dunn, Hildegarde Heymann. The influence of alcohol on the sensory perception of red wines. Food Quality and Preference, 2013, 28, 235-243.
Best Regards

Reviewer 2 Report
Manuscript ID: beverages-2036060
Title: Sparkling cider paired with Italian cheese: sensory analysis and consumer assessment
The proposed manuscript ID beverages-2036060 is about sensory analysis and consumer evaluation of sparkling cider in combination with the consumption of Italian cheese.
The authors created an experiment in a way that trained panel and consumers to evaluate both, Italian cheese and sparkling ciders and their combination by applying a hedonic scale.
In a conclusion, the authors state that the consumption of cheese before drinking sparkling cider significantly changes the sensory profile of the sparkling cider. Futher, Italian cheese and sparking cider could be combined.
The manuscript is well-written and scientifically sound. But some omissions are noticed which are emphasized in the comments highlighted in the document.
According to the subject of this manuscript, I found it interesting and I think it would be very interesting for readers, too.
Great job!
Best regards!

Author Response
Dear Reviewer
We have made the corrections, as suggested.
- Free line could be deleted
We have deleted the free line of Table 2
- Font is different line 194-199
We have corrected in the text, as follow:
Data were statistically processed by Excel (Microsoft Corporation, Redmond, USA), Statgraphics Centurion XVI (StatPoint Technologies Inc., Warrenton, USA) and OriginPro (OriginLab Corporation, Northampton, USA). A descriptive statistical study, spider plots (sensory parameters) was conducted. Analysis of variance (ANOVA) followed by Tukey’s test and HSD were applied for inferential study (p <0.05). Principal component analysis (PCA) was applied to identify sensory parameters, ciders and cider-cheese paired.
- Table title should be above the table, line 255-257.
We have corrected, as suggested
- Line 433-444, Figure 3 A and 3 B:
We have corrected the Figure 3, by adding 3A and 3B.
- Conclusion
We added as follow:
In particular, the reduction in the perception of astringency, after tasted the cheeses, is more evident in the ciders in which this parameter is more expressed; moreover, astringency and bitterness are always opposite to the assessment. This means that disliked characteristics in the cider can be modulated in the pairing, and result in an enhanced preference for the cider.
If the effervescence is considered, it is attenuated by the fat component of the cheeses, in all cases.
As regards the olfactory intensity, it is not clear whether the parameter is due to an enhancement effect linked to the pairing or to the strong odor component of the cheeses which in many cases dominates that of the ciders. In general, the sapidity and umami taste, developed during the aging of the cheeses, may have influenced the sensory profile of the pairings compared to the notes of milk and sweetness, typical of fresh cheeses, which are more similar to the sensory profile of cider, tested alone.
Best Regards

Reviewer 3 Report
the paper is well structured, well written and presented. I got throught it very fast and I have found it very interesting.
The topic has limited interest in Italian market, but the study could present relevance by looking at future possible developments of the market, nowaday rather limited.
Method is appropriate although the types of cider selected for the analyses are not well explainer, nor the process of selection. How many ciders in italy? why those 6? This sould be imporved to give more scientific sound to the paper.
Author Response
Dear Reviewer
We have made the correctios, as suggested.
Comments and Suggestions for Authors
Method is appropriate although the types of cider selected for the analyses are not well explainer, nor the process of selection. How many ciders in italy? why those 6? This sould be imporved to give more scientific sound to the paper.
- How many ciders in italy?
According to the reviewer, we have explained the Italian tradition in cider production.
In the Italian market, the production and marketing of apple cider, or generally fermented fruit other than grapes, is not widespread. Some cider-growing realities are present on the Italian national territory but are essentially aimed at an artisanal production linked to ancient traditions.
As far as sider-making is concerned, the areas are concentrated in northern Italy, where cider has a tradition: Piedmont, Trentino Alto-Adige, Friuli Venezia Giulia and Val d'Aosta. The productions of these Italian regions, however, are more limited than those of French, especially in Normandy which holds the record, and of the Spanish or English ones.
To hinder its diffusion, there is above all the competition from wine, the most deeply rooted alcoholic beverage in Italy. Today, however, in Italy cider production companies are once again appearing on the market. Furthermore, this product could solve the problem of recovering certain mountain areas, in which to enhance the typical production which can constitute a good source of income in an agricultural economy. The search for a subtle balance between the typical organoleptic aspect, the production technologies and the high naturalness that distinguish this beverage could certainly favor the approach of potential consumers. [https://sidrodimele.it/]
why those 6?
These 6 Italian sparkling ciders were chosen because they are known in the production areas and marketed in Italy; moreover, although they belong to the sparkling category, they have different sensory characteristics, linked to the tradition (apple variety and cider production techniques) of the production area. Even if there are other areas and types of ciders in Italy, the study is preliminary and could represent the beginning for exploring the pairing of ciders with other typical cheese and/or foods.
Best Regards
